# Matrix Completion Under Monotonic Single Index Models

**Ravi Ganti**
Wisconsin Institutes for Discovery
UW-Madison
gantimahapat@wisc.edu

**Laura Balzano**
Electrical Engineering and Computer Sciences
University of Michigan Ann Arbor
girasole@umich.edu

**Rebecca Willett**
Department of Electrical and Computer Engineering
UW-Madison
rmwillett@wisc.edu

## Abstract

Most recent results in matrix completion assume that the matrix under consideration is low-rank or that the columns are in a union of low-rank subspaces. In real-world settings, however, the linear structure underlying these models is distorted by a (typically unknown) nonlinear transformation. This paper addresses the challenge of matrix completion in the face of such nonlinearities. Given a few observations of a matrix that are obtained by applying a Lipschitz, monotonic function to a low rank matrix, our task is to estimate the remaining unobserved entries. We propose a novel matrix completion method that alternates between low-rank matrix estimation and monotonic function estimation to estimate the missing matrix elements. Mean squared error bounds provide insight into how well the matrix can be estimated based on the size, rank of the matrix and properties of the nonlinear transformation. Empirical results on synthetic and real-world datasets demonstrate the competitiveness of the proposed approach.

## 1 Introduction

In matrix completion, one has access to a matrix with only a few observed entries, and the task is to estimate the entire matrix using the observed entries. This problem has a plethora of applications such as collaborative filtering, recommender systems [1] and sensor networks [2]. Matrix completion has been well studied in machine learning, and we now know how to recover certain matrices given a few observed entries of the matrix [3, 4] when it is assumed to be low rank. Typical work in matrix completion assumes that the matrix to be recovered is incoherent, low rank, and entries are sampled uniformly at random [5, 6, 4, 3, 7, 8]. While recent work has focused on relaxing the incoherence and sampling conditions under which matrix completion succeeds, there has been little work for matrix completion when the underlying matrix is of high rank. More specifically, we shall assume that the matrix that we need to complete is obtained by applying some unknown, non-linear function to each element of an unknown low-rank matrix. Because of the application of a non-linear transformation, the resulting ratings matrix tends to have a large rank. To understand the effect of the application of non-linear transformation on a low-rank matrix, we shall consider the following simple experiment: Given an $n \times m$ matrix $X$, let $X = \sum_{i=1}^{m} \sigma_i u_i v_i^{\top}$ be its SVD. The rank of the matrix $X$ is the number of non-zero singular values. Given an $\epsilon \in (0,1)$, define the effective rank of $X$ as follows:

$$r_\epsilon(X) = \min \left\{ k \in \mathbb{N} : \sqrt{\frac{\sum_{j=k+1}^{m} \sigma_j^2}{\sum_{j=1}^{m} \sigma_j^2}} \leq \epsilon \right\}. \qquad (1)$$

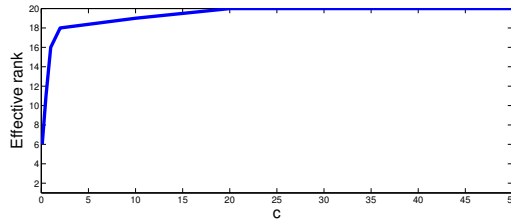

Figure 1: The plot shows the $r_{0.01}(X)$ defined in equation (1) obtained by applying a non-linear function $g^\star$ to each element of $Z$, where $g^\star(z) = \frac{1}{1+\exp(-cz)}$. $Z$ is a $30 \times 20$ matrix of rank 5.

The effective rank of $X$ tells us the rank $k$ of the lowest rank approximator $\hat{X}$ that satisfies

$$\frac{||\hat{X} - X||_F}{||X||_F} \leq \epsilon. \tag{2}$$

In figure (1), we show the effect of applying a non-linear monotonic function $g^\star(z) = \frac{1}{1+\exp(-cz)}$ to the elements of a low-rank matrix $Z$. As $c$ increases both the rank of $X$ and its effective rank $r_\epsilon(X)$ grow rapidly with $c$, rendering traditional matrix completion methods ineffective even in the presence of mild nonlinearities.

## 1.1 Our Model and contributions

In this paper we consider the high-rank matrix completion problem where the data generating process is as follows: There is some unknown matrix $Z^\star \in \mathbb{R}^{n \times m}$ with $m \leq n$ and of rank $r \ll m$. A non-linear, monotonic, $L$- Lipschitz function $g^\star$ is applied to each element of the matrix $Z^\star$ to get another matrix $M^\star$. A noisy version of $M^\star$, which we call $X$, is observed on a subset of indices denoted by $\Omega \subset [n] \times [m]$.

$$M^\star_{i,j} = g^\star(Z^\star_{i,j}), \ \forall i \in [n], j \in [m] \tag{3}$$

$$X_\Omega = (M^\star + N)_\Omega \tag{4}$$

The function $g^\star$ is called the transfer function. We shall assume that $\mathbb{E}[N] = 0$, and the entries of $N$ are i.i.d. We shall also assume that the index set $\Omega$ is generated uniformly at random with replacement from the set $[n] \times [m]$ [1]. Our task is to reliably estimate the entire matrix $M^\star$ given observations of $X$ on $\Omega$. We shall call the above model as Monotonic Matrix Completion (MMC). To illustrate our framework we shall consider the following two simple examples. In recommender systems users are required to provide discrete ratings of various objects. For example, in the Netflix problem users are required to rate movies on a scale of $1 - 5$ [2]. These discrete scores can be thought of as obtained by applying a rounding function to some ideal real valued score matrix given by the users. This real-valued score matrix may be well modeled by a low-rank matrix, but the application of the rounding function [3] increases the rank of the original low-rank matrix. Another important example is that of completion of Gaussian kernel matrices. Gaussian kernel matrices are used in kernel based learning methods. The Gaussian kernel matrix of a set of $n$ points is an $n \times n$ matrix obtained by applying the Gaussian function on an underlying Euclidean distance matrix. The Euclidean distance matrix is a low-rank matrix [9]. However, in many cases one cannot measure all pair-wise distances between objects, resulting in an incomplete Euclidean distance matrix and hence an incomplete kernel matrix. Completing the kernel matrix can then be viewed as completing a matrix of large rank.

In this paper we study this matrix completion problem and provide algorithms with provable error guarantees. Our contributions are as follows:

1. In Section (3) we propose an optimization formulation to estimate matrices in the above described context. In order to do this we introduce two formulations, one using a squared

loss, which we call MMC - LS, and another using a calibrated loss function, which we call as MMC $- c$. For both these formulations we minimize w.r.t. $M^\star$ and $g^\star$. This calibrated loss function has the property that the minimizer of the calibrated loss satisfies equation (3).

2. We propose alternating minimization algorithms to solve our optimization problem. Our proposed algorithms, called MMC $- c$ and MMC-LS, alternate between solving a quadratic program to estimate $g^\star$ and performing projected gradient descent updates to estimate the matrix $Z^\star$. MMC outputs the matrix $\hat{M}$ where $\hat{M}_{i,j} = \hat{g}(\hat{Z}_{i,j})$.

3. In Section (4) we analyze the mean squared error (MSE) of the matrix $\hat{M}$ returned by one step of the MMC $- c$ algorithm. The upper bound on the MSE of the matrix $\hat{M}$ output by MMC depends only on the rank $r$ of the matrix $Z^\star$ and not on the rank of matrix $M^\star$. This property makes our analysis useful because the matrix $M^\star$ could be potentially high rank and our results imply reliable estimation of a high rank matrix with error guarantees that depend on the rank of the matrix $Z^\star$.

4. We compare our proposed algorithms to state-of-art implementations of low rank matrix completion on both synthetic and real datasets (Section 5).

## 2    Related work

Classical matrix completion with and without noise has been investigated by several authors [5, 6, 4, 3, 7, 8]. The recovery techniques proposed in these papers solve a convex optimization problem that minimizes the nuclear norm of the matrix subject to convex constraints. Progress has also been made on designing efficient algorithms to solve the ensuing convex optimization problem [10, 11, 12, 13]. Recovery techniques based on nuclear norm minimization guarantee matrix recovery under the condition that a) the matrix is low rank, b) the matrix is incoherent or not very spiky, and c) the entries are observed uniformly at random. Literature on high rank matrix completion is relatively sparse. When columns or rows of the matrix belong to a union of subspaces, then the matrix tends to be of high rank. For such high rank matrix completion problems algorithms have been proposed that exploit the fact that multiple low-rank subspaces can be learned by clustering the columns or rows and learning subspaces from each of the clusters. While Eriksson et al. [14] suggested looking at the neighbourhood of each incomplete point for completion, [15] used a combination of spectral clustering techniques as done in [16, 17] along with learning sparse representations via convex optimization to estimate the incomplete matrix. Singh et al. [18] consider a certain specific class of high-rank matrices that are obtained from ultra-metrics. In [19] the authors consider a model similar to ours, but instead of learning a single monotonic function, they learn multiple monotonic functions, one for each row of the matrix. However, unlike in this paper, their focus is on a ranking problem and their proposed algorithms lack theoretical guarantees.

Davenport et al [20] studied the one-bit matrix completion problem. Their model is a special case of the matrix completion model considered in this paper. In the one-bit matrix completion problem we assume that $g^\star$ is known and is the CDF of an appropriate probability distribution, and the matrix $X$ is a boolean matrix where each entry takes the value 1 with probability $M_{i,j}$, and 0 with probability $1 - M_{i,j}$. Since $g^\star$ is known, the focus in one-bit matrix completion problems is accurate estimation of $Z^\star$.

To the best of our knowledge the MMC model considered in this paper has not been investigated before. The MMC model is inspired by the single-index model (SIM) that has been studied both in statistics [21, 22] and econometrics for regression problems [23, 24]. Our MMC model can be thought of as an extension of SIM to matrix completion problems.

## 3    Algorithms for matrix completion

Our goal is to estimate $g^\star$ and $Z^\star$ from the model in equations (3- 4). We approach this problem via mathematical optimization. Before we discuss our algorithms, we mention in brief an algorithm for the problem of learning Lipschitz, monotonic functions in 1- dimension. This algorithm will be used for learning the link function in MMC.

**The LPAV algorithm:** Suppose we are given data $(p_1, y_1), \ldots (p_n, y_n)$, where $p_1 \leq p_2 \ldots \leq p_n$, and $y_1, \ldots, y_n$ are real numbers. Let $\mathcal{G} \stackrel{\text{def}}{=} \{g : \mathbb{R} \to \mathbb{R}, \ g \text{ is L-Lipschitz and monotonic}\}$. The LPAV [4] algorithm introduced in [21] outputs the best function $\hat{g}$ in $\mathcal{G}$ that minimizes $\sum_{i=1}^{n}(g(p_i) - y_i)^2$. In order to do this, the LPAV first solves the following optimization problem:

$$\hat{z} = \arg \min_{z \in \mathbb{R}^n} \ \|z - y\|_2^2 \quad \textbf{s.t.} \ \ 0 \leq z_j - z_i \leq L(p_j - p_i) \text{ if } p_i \leq p_j \tag{5}$$

where, $\hat{g}(p_i) \stackrel{\text{def}}{=} \hat{z}_i$. This gives us the value of $\hat{g}$ on a discrete set of points $p_1, \ldots, p_n$. To get $\hat{g}$ everywhere else on the real line, we simply perform linear interpolation as follows:

$$\hat{g}(\zeta) = \begin{cases} \hat{z}_1, & \text{if } \zeta \leq p_1 \\ \hat{z}_n, & \text{if } \zeta \geq p_n \\ \mu\hat{z}_i + (1 - \mu)\hat{z}_{i+1} & \text{if } \zeta = \mu p_i + (1 - \mu)p_{i+1} \end{cases} \tag{6}$$

## 3.1 Squared loss minimization

A natural approach to the monotonic matrix completion problem is to learn $g^\star, Z^\star$ via squared loss minimization. In order to do this we need to solve the following optimization problem:

$$\min_{g,Z} \sum_{\Omega} (g(Z_{i,j}) - X_{i,j})^2$$
$$g : \mathbb{R} \to \mathbb{R} \text{ is L-Lipschitz and monotonic} \tag{7}$$
$$rank(Z) \leq r.$$

The problem is a non-convex optimization problem individually in parameters $g, Z$. A reasonable approach to solve this optimization problem would be to perform optimization w.r.t. each variable while keeping the other variable fixed. For instance, in iteration $t$, while estimating $Z$ one would keep $g$ fixed, to say $g^{t-1}$, and then perform projected gradient descent w.r.t. $Z$. This leads to the following updates for $Z$:

$$Z_{i,j}^t \leftarrow Z_{i,j}^{t-1} - \eta(g^{t-1}(Z_{i,j}^{t-1}) - X_{i,j})(g^{t-1})'(Z_{i,j}^{t-1}) , \forall (i,j) \in \Omega \tag{8}$$
$$Z^t \leftarrow P_r(Z^t) \tag{9}$$

where $\eta > 0$ is a step-size used in our projected gradient descent procedure, and $P_r$ is projection on the rank $r$ cone. The above update involves both the function $g^{t-1}$ and its derivative $(g^{t-1})'$. Since our link function is monotonic, one can use the LPAV algorithm to estimate this link function $g^{t-1}$. Furthermore since LPAV estimates $g^{t-1}$ as a piece-wise linear function, the function has a sub-differential everywhere and the sub-differential $(g^{t-1})'$ can be obtained very cheaply. Hence, the projected gradient update shown in equation (8) along with the LPAV algorithm can be iteratively used to learn estimates for $Z^\star$ and $g^\star$. We shall call this algorithm as MMC$-$LS. Incorrect estimation of $g^{t-1}$ will also lead to incorrect estimation of the derivative $(g^{t-1})'$. Hence, we would expect MMC$-$LS to be less accurate than a learning algorithm that does not have to estimate $(g^{t-1})'$. We next outline an approach that provides a principled way to derive updates for $Z^t$ and $g^t$ that does not require us to estimate derivatives of the transfer function, as in MMC$-$LS.

## 3.2 Minimization of a calibrated loss function and the MMC algorithm.

Let $\Phi : \mathbb{R} \to \mathbb{R}$ be a differentiable function that satisfies $\Phi' = g^\star$. Furthermore, since $g^\star$ is a monotonic function, $\Phi$ will be a convex loss function. Now, suppose $g^\star$ (and hence $\Phi$) is known. Consider the following function of $Z$

$$\mathcal{L}(Z; \Phi, \Omega) = \mathbb{E}_X \left( \sum_{(i,j) \in \Omega} \Phi(Z_{i,j}) - X_{i,j}Z_{i,j} \right). \tag{10}$$

The above loss function is convex in $Z$, since $\Phi$ is convex. Differentiating the expression on the R.H.S. of Equation 10 w.r.t. $Z$, and setting it to 0, we get

$$\sum_{(i,j) \in \Omega} g^\star(Z_{i,j}) - \mathbb{E}X_{i,j} = 0. \tag{11}$$

The MMC model shown in Equation (3) satisfies Equation (11) and is therefore a minimizer of the loss function $\mathcal{L}(Z; \Phi, \Omega)$. Hence, the loss function (10) is "calibrated" for the MMC model that we are interested in. The idea of using calibrated loss functions was first introduced for learning single index models [25]. When the transfer function is identity, $\Phi$ is a quadratic function and we get the squared loss approach that we discussed in section (3.1).

The above discussion assumes that $g^\star$ is known. However in the MMC model this is not the case. To get around this problem, we consider the following optimization problem

$$\min_{\Phi, Z} \mathcal{L}(\Phi, Z; \Omega) = \min_{\Phi, Z} \mathbb{E}_X \sum_{(i,j) \in \Omega} \Phi(Z_{i,j}) - X_{i,j} Z_{i,j} \tag{12}$$

where $\Phi : \mathbb{R} \to \mathbb{R}$ is a convex function, with $\Phi' = g$ and $Z \in \mathbb{R}^{m \times n}$ is a low-rank matrix. Since, we know that $g^\star$ is a Lipschitz, monotonic function, we shall solve a constrained optimization problem that enforces Lipschitz constraints on $g$ and low rank constraints on $Z$. We consider the sample version of the optimization problem shown in equation (12).

$$\min_{\substack{\Phi \\ rank(Z) \leq r}} \mathcal{L}(\Phi, Z; \Omega) = \min_{\Phi, Z} \sum_{(i,j) \in \Omega} \Phi(Z_{i,j}) - X_{i,j} Z_{i,j} \tag{13}$$

The pseudo-code of our algorithm MMC that solves the above optimization problem (13) is shown in algorithm (1). MMC optimizes for $\Phi$ and $Z$ alternatively, where we fix one variable and update another.

At the start of iteration $t$, we have at our disposal iterates $\hat{g}^{t-1}$, and $Z^{t-1}$. To update our estimate of $Z$, we perform gradient descent with fixed $\Phi$ such that $\Phi' = \hat{g}^{t-1}$. Notice that the objective in equation (13) is convex w.r.t. $Z$. This is in contrast to the least squares objective where the objective in equation (7) is non-convex w.r.t. $Z$. The gradient of $\mathcal{L}(Z; \Phi)$ w.r.t. $Z$ is

$$\nabla_{Z_{i,j}} \mathcal{L}(Z; \Phi) = \sum_{(i,j) \in \Omega} \hat{g}^{t-1}(\hat{Z}_{i,j}^{t-1}) - X_{i,j}. \tag{14}$$

Gradient descent updates on $\hat{Z}^{t-1}$ using the above gradient calculation leads to an update of the form

$$\hat{Z}_{i,j}^t \leftarrow \hat{Z}_{i,j}^{t-1} - \eta(\hat{g}^{t-1}(\hat{Z}_{i,j}^{t-1}) - X_{i,j}) \mathbb{1}_{(i,j) \in \Omega}$$
$$\hat{Z}^t \leftarrow \mathcal{P}_r(\hat{Z}^t) \tag{15}$$

Equation (15) projects matrix $\hat{Z}^t$ onto a cone of matrices of rank $r$. This entails performing SVD on $\hat{Z}^t$ and retaining the top $r$ singular vectors and singular values while discarding the rest. This is done in steps 4, 5 of Algorithm (1). As can be seen from the above equation we *do not* need to estimate derivative of $\hat{g}^{t-1}$. This, along with the convexity of the optimization problem in Equation (13) w.r.t. $Z$ for a given $\Phi$ are two of the key advantages of using a calibrated loss function over the previously proposed squared loss minimization formulation.

**Optimization over $\Phi$.** In round $t$ of algorithm (1), we have $\hat{Z}^t$ after performing steps 4, 5. Differentiating the objective function in equation (13) w.r.t. $Z$, we get that the optimal $\Phi$ function should satisfy

$$\sum_{(i,j) \in \Omega} \hat{g}^t(\hat{Z}_{i,j}^t) - X_{i,j} = 0, \tag{16}$$

where $\Phi' = \hat{g}^t$. This provides us with a strategy to calculate $\hat{g}^t$. Let, $\hat{X}_{i,j} \stackrel{\text{def}}{=} \hat{g}^t(\hat{Z}_{i,j}^t)$. Then solving the optimization problem in equation (16) is equivalent to solving the following optimization problem.

$$\min_{\hat{X}} \sum_{(i,j) \in \Omega} (\hat{X}_{i,j} - X_{i,j})^2 \tag{17}$$
$$\text{subject to: } 0 \leq -\hat{X}_{i,j} + \hat{X}_{k,l} \leq L(\hat{Z}_{k,l}^t - \hat{Z}_{i,j}^t) \text{ if } \hat{Z}_{i,j}^t \leq \hat{Z}_{k,l}^t, \ (i,j) \in \Omega, (k,l) \in \Omega$$

where $L$ is the Lipschitz constant of $g^\star$. We shall assume that $L$ is known and does not need to be estimated. The gradient, w.r.t. $\hat{X}$, of the objective function, in equation (17), when set to zero is

the same as Equation (16). The constraints enforce monotonicity of $\hat{g}^t$ and the Lipschitz property of $\hat{g}^t$. The above optimization routine is exactly the LPAV algorithm. The solution $\hat{X}$ obtained from solving the LPAV problem can be used to define $\hat{g}^t$ on $X_\Omega$. These two steps are repeated for $T$ iterations. After $T$ iterations we have $\hat{g}^T$ defined on $\hat{Z}_\Omega^T$. In order to define $\hat{g}^T$ everywhere else on the real line we perform linear interpolation as shown in equation (6).

---

**Algorithm 1** Monotonic Matrix Completion (MMC)

---

**Input:** Parameters $\eta > 0, T > 0, r$, Data:$X_\Omega, \Omega$
**Output:** $\hat{M} = \hat{g}^T(\hat{Z}^T)$
 1: Initialize $\hat{Z}^0 = \frac{mn}{|\Omega|}X_\Omega$, where $X_\Omega$ is the matrix $X$ with zeros filled in at the unobserved locations.
 2: Initialize $\hat{g}^0(z) = \frac{|\Omega|}{mn}z$
 3: **for** $t = 1, \ldots, T$ **do**
 4: $\quad$ $\hat{Z}_{i,j}^t \leftarrow \hat{Z}_{i,j}^{t-1} - \eta(\hat{g}^{t-1}(\hat{Z}_{i,j}^{t-1}) - X_{i,j})\mathbb{1}_{(i,j)\in\Omega}$
 5: $\quad$ $\hat{Z}^t \leftarrow \mathcal{P}_r(\hat{Z}^t)$
 6: $\quad$ Solve the optimization problem in (17) to get $\hat{X}$
 7: $\quad$ Set $\hat{g}^t(\hat{Z}_{i,j}^t) = \hat{X}_{i,j}$ for all $(i,j) \in \Omega$.
 8: **end for**
 9: Obtain $\hat{g}^T$ on the entire real line using linear interpolation shown in equation (6).

---

Let us now explain our initialization procedure. Define $X_\Omega \overset{\text{def}}{=} \sum_{j=1}^{|\Omega|} X \circ \Delta_j$, where each $\Delta_j$ is a boolean mask with zeros everywhere and a 1 at an index corresponding to the index of an observed entry. $A \circ B$ is the Hadamard product, i.e. entry-wise product of matrices $A, B$. We have $|\Omega|$ such boolean masks each corresponding to an observed entry. We initialize $\hat{Z}_\Omega^0$ to $\frac{mn}{|\Omega|}X_\Omega = \frac{mn}{|\Omega|}\sum_{j=1}^{|\Omega|} X \circ \Delta_j$. Because each observed index is assumed to be sampled uniformly at random with replacement, our initialization is guaranteed to be an unbiased estimate of $X$.

# 4 MSE Analysis of MMC

We shall analyze our algorithm, MMC, for the case of $T = 1$, under the modeling assumption shown in Equations (4) and (3). Additionally, we will assume that the matrices $Z^\star$ and $M^\star$ are bounded entry-wise in absolute value by 1. When $T = 1$, the MMC algorithm estimates $\hat{Z}$, $\hat{g}$ and $\hat{M}$ as follows

$$\hat{Z} = \mathcal{P}_r\left(\frac{mnX_\Omega}{|\Omega|}\right). \tag{18}$$

$\hat{g}$ is obtained by solving the LPAV problem from Equation (17) with $\hat{Z}$ shown in Equation (18). This allows us to define $\hat{M}_{i,j} = \hat{g}(\hat{Z}_{i,j}), \forall i = [n], j = [m]$.

Define the mean squared error (MSE) of our estimate $\hat{M}$ as

$$MSE(\hat{M}) = \mathbb{E}\left[\frac{1}{mn}\sum_{i=1}^{n}\sum_{j=1}^{m}(\hat{M}_{i,j} - M_{i,j})^2\right]. \tag{19}$$

Denote by $||M||$ the spectral norm of a matrix $M$. We need the following additional technical assumptions:

A1. $\|Z^\star\| = O(\sqrt{n})$.

A2. $\sigma_{r+1}(X) = \tilde{O}(\sqrt{n})$ with probability at least $1 - \delta$, where $\tilde{O}$ hides terms logarithmic in $1/\delta$.

$Z^\star$ has entries bounded in absolute value by 1. This means that in the worst case, $||Z^\star|| = \sqrt{mn}$. Assumption A1 requires that the spectral norm of $Z^\star$ is not very large. Assumption A2 is a weak assumption on the decay of the spectrum of $M^\star$. By assumption $X = M^\star + N$. Applying Weyl's

inequality we get $\sigma_{r+1}(X) \leq \sigma_{r+1}(M^\star) + \sigma_1(N)$. Since $N$ is a zero-mean noise matrix with independent bounded entries, $N$ is a matrix with sub-Gaussian entries. This means that $\sigma_1(N) = \tilde{O}(\sqrt{n})$ with high probability. Hence, assumption A2 can be interpreted as imposing the condition $\sigma_{r+1}(M^\star) = O(\sqrt{n})$. This means that while $M^\star$ could be full rank, the $(r+1)^{\text{th}}$ singular value of $M^\star$ cannot be too large.

**Theorem 1.** *Let* $\mu_1 \stackrel{\text{def}}{=} \mathbb{E}||N||, \mu_2 \stackrel{\text{def}}{=} \mathbb{E}||N||^2$. *Let* $\alpha = ||M^\star - Z^\star||$. *Then, under assumptions A1 and A2, the MSE of the estimator output by MMC with $T = 1$ is given by*

$$MSE(\hat{M}) = O\left( \sqrt{\frac{r}{m}} + \frac{\sqrt{mn\log(n)}}{|\Omega|} + \frac{mn}{|\Omega|^{3/2}} + \sqrt{\frac{r}{m\sqrt{n}}\left(\mu_1 + \frac{\mu_2}{\sqrt{n}}\right)} + \right.$$
$$\left. \sqrt{\frac{r\alpha}{m\sqrt{n}}\left(1 + \frac{\alpha}{\sqrt{n}}\right)} + \sqrt{\frac{rmn\log^2(n)}{|\Omega|^2}} \right). \tag{20}$$

where $O(\cdot)$ notation hides universal constants, and the Lipschitz constant $L$ of $g^\star$. We would like to mention that the result derived for MMC-1 can be made to hold true for $T > 1$, by an additional large deviation argument.

**Interpretation of our results:** Our upper bounds on the MSE of MMC depends on the quantity $\alpha = ||M^\star - Z^\star||$, and $\mu_1, \mu_2$. Since matrix $N$ has independent zero-mean entries which are bounded in absolute value by 1, $N$ is a sub-Gaussian matrix with independent entries. For such matrices $\mu_1 = O(\sqrt{n}), \mu_2 = O(n)$ (see Theorem 5.39 in [26]). With these settings we can simplify the expression in Equation (20) to

$$MSE(\hat{M}) = \tilde{O}\left( \sqrt{\frac{r}{m}} + \frac{\sqrt{mn\log(n)}}{|\Omega|} + \frac{mn}{|\Omega|^{3/2}} + \sqrt{\frac{r\alpha}{m\sqrt{n}}\left(1 + \frac{\alpha}{\sqrt{n}}\right)} + \sqrt{\frac{rmn\log^2(n)}{|\Omega|^2}} \right).$$

A remarkable fact about our sample complexity results is that the sample complexity is independent of the rank of matrix $M^\star$, which could be large. Instead it depends on the rank of matrix $Z^\star$ which we assume to be small. The dependence on $M^\star$ is via the term $\alpha = ||M^\star - Z^\star||$. From equation (4) it is evident that the best error guarantees are obtained when $\alpha = O(\sqrt{n})$. For such values of $\alpha$ equation (4) reduces to,

$$MSE(\hat{M}) = \tilde{O}\left( \sqrt{\frac{r}{m}} + \frac{\sqrt{mn\log(n)}}{|\Omega|} + \frac{mn}{|\Omega|^{3/2}} + \frac{\sqrt{mn}}{|\Omega|} + \sqrt{\frac{rmn\log^2(n)}{|\Omega|^2}} \right).$$

This result can be converted into a sample complexity bound as follows. If we are given $|\Omega| = \tilde{O}\left(\left(\frac{mn}{\epsilon}\right)^{2/3}\right)$, then $MSE(\hat{M}) \leq \sqrt{\frac{r}{m}} + \epsilon$. It is important to note that the floor of the MSE is $\sqrt{\frac{r}{m}}$, which depends on the rank of $Z^\star$ and not on $rank(M^\star)$, which can be much larger than $r$.

## 5 Experimental results

We compare the performance of $MMC-1$, $MMC-c$, MMC- LS, and nuclear norm based low-rank matrix completion (LRMC) [4] on various synthetic and real world datasets. The objective metric that we use to compare different algorithms is the root mean squared error (RMSE) of the algorithms on unobserved, test indices of the incomplete matrix.

### 5.1 Synthetic experiments

For our synthetic experiments we generated a random $30 \times 20$ matrix $Z^\star$ of rank 5 by taking the product of two random Gaussian matrices of size $n \times r$, and $r \times m$, with $n = 30, m = 20, r = 5$. The matrix $M^\star$ was generated using the function, $g^\star(M^\star_{i,j}) = 1/(1 + \exp(-cZ^\star_{i,j}))$, where $c > 0$. By increasing $c$, we increase the Lipschitz constant of the function $g^\star$, making the matrix completion task harder. For large enough $c$, $M_{i,j} \approx \text{sgn}(Z_{i,j})$. We consider the noiseless version of the problem where $X = M^\star$. Each entry in the matrix $X$ was sampled with probability $p$, and the sampled entries are observed. This makes $\mathbb{E}|\Omega| = mnp$. For our implementations we assume that $r$ is unknown,

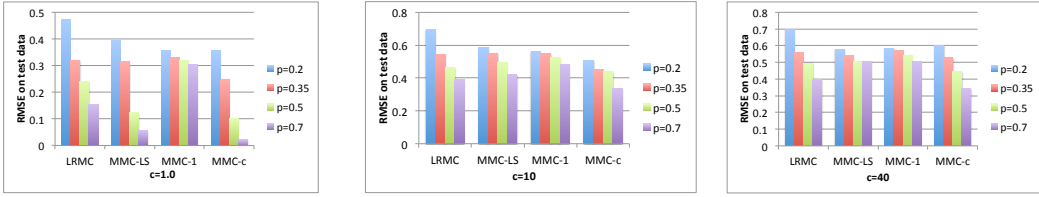

Figure 2: RMSE of different methods at different values of $c$.

and estimate it either (i) via the use of a dedicated validation set in the case of MMC $-1$ or (ii) adaptively, where we progressively increase the estimate of our rank until a sufficient decrease in error over the training set is achieved [13]. For an implementation of the LRMC algorithm we used a standard off-the-shelf implementation from TFOCS [27]. In order to speed up the run time of MMC, we also keep track of the training set error, and terminate iterations if the relative residual on the training set goes below a certain threshold [5]. In the supplement we provide a plot that demonstrates that, for MMC $- c$, the RMSE on the training dataset has a decreasing trend and reaches the required threshold in at most 50 iterations. Hence, we set $T = 50$. Figure (2) show the RMSE of each method for different values of $p, c$. As one can see from figure (2), the RMSE of all the methods improves for any given $c$ as $p$ increases. This is expected since as $p$ increases $\mathbb{E}|\Omega| = pmn$ also increases. As $c$ increases, $g^\star$ becomes steeper increasing the effective rank of $X$. This makes matrix completion task hard. For small $p$, such as $p = 0.2$, MMC $- 1$ is competitive with MMC $- c$ and MMC$-$LS and is often the best. In fact for small $p$, irrespective of the value of $c$, LRMC is far inferior to other methods. For larger $p$, MMC $- c$ works the best achieving smaller RMSE over other methods.

## 5.2 Experiments on real datasets

We performed experimental comparisons on four real world datasets: paper recommendation, Jester-3, ML-100k, Cameraman. All of the above datasets, except the Cameraman dataset, are ratings datasets, where users have rated a few of the several different items. For the Jester-3 dataset we used 5 randomly chosen ratings for each user for training, 5 randomly chosen rating for validation and the remaining for testing. ML-100k comes with its own training and testing dataset. We used 20% of the training data for validation. For the Cameraman and the paper recommendation datasets 20% of the data was used for training, 20% for validation and the rest for testing. The baseline algorithm chosen for low rank matrix completion is LMaFit-A [13] [6].

For each of the datasets we report the RMSE of MMC $- 1$, MMC $- c$, and LMaFit-A on the test sets. We excluded MMC-LS from these experiments because in all of our datasets the number of observed entries is a very small fraction of the total number of entries, and from our results on synthetic datasets we know that MMC$-$ LS is not the best performing algorithm in such cases. Table 1 shows the RMSE over the test set of the different matrix completion methods. As we see the RMSE of MMC $- c$ is the smallest of all the methods, surpassing LMaFit-A by a large margin.

Table 1: RMSE of different methods on real datasets.

| Dataset | Dimensions | $|\Omega|$ | $r_{0.01}(X)$ | LMaFit-A | MMC $- 1$ | MMC $- c$ |
|---|---|---|---|---|---|---|
| PaperReco | $3426 \times 50$ | 34294 | 47 | 0.4026 | 0.4247 | **0.2965** |
| Jester-3 | $24938 \times 100$ | 124690 | 66 | 6.8728 | 5.327 | **5.2348** |
| ML-100k | $1682 \times 943$ | 64000 | 391 | 3.3101 | 1.388 | **1.1533** |
| Cameraman | $1536 \times 512$ | 157016 | 393 | 0.0754 | 0.1656 | **0.06885** |

## 6 Conclusions and future work

We have investigated a new framework and for high rank matrix completion problems called monotonic matrix completion and proposed new algorithms. In the future we would like to investigate if one could relax improve the theoretical results.

## Footnotes

[1]By $[n]$ we denote the set $\{1, 2 \ldots, n\}$

[2]This is typical of many other recommender engines such as Pandora.com, Last.fm and Amazon.com.

[3]Technically the rounding function is not a Lipschitz function but can be well approximated by a Lipschitz function.

[4] LPAV stands for Lipschitz Pool Adjacent Violator

[5]For our experiments this threshold is set to 0.001.

[6]http://lmafit.blogs.rice.edu/. The parameter $k$ in the LMaFit algorithm was set to effective rank, and we used est_rank=1 for LMaFit-A.

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
