[Supplementary Material]

# Supplement:Matrix Completion Under Monotonic Single Index Models

## 1  Error Analysis of Monotonic Matrix Completion

For our technical results, we shall consider a simpler easier-to-anlyze sampling model. We shall assume that instead of choosing each entry of the matrix independently with some probability, we instead choose $|\Omega|$ number of entries in the matrix independently with replacement. We shall analyze our algorithm, $MMC$, for the case of $T = 1$. When $T = 1$, we get

$$\hat{Z} = P_r\left(\frac{mnX_\Omega}{|\Omega|}\right) \tag{1}$$

$$\hat{g} = LPAV(\hat{Z}_\Omega, X_\Omega) \tag{2}$$

$$\hat{M}_{i,j} = \hat{g}(\hat{Z}_{i,j}), \forall i = [m], j = [n], \tag{3}$$

For technical convenience, let $p = \frac{1}{mn}$. Note that this $p$ is not the same as $p$ used in the main paper. Here, $p = \frac{1}{mn}$, is defined keeping in mind that $\frac{|\Omega|}{mn}$ can be roughly thought of as the probability of sampling an element in the matrix. Finally, define the mean squared error (MSE) of our estimate $\hat{M}$ can be defined as

$$MSE(\hat{M}) = \mathbb{E}\left[\frac{1}{mn}\sum_{i=1}^{n}\sum_{j=1}^{m}(\hat{M}_{i,j} - M_{i,j})^2\right]. \tag{4}$$

We are interested in analyzing the $MSE$ of $\hat{M}$ output by $MMC$ for $T = 1$. We shall make the following assumptions

## 2  MMC model and technical assumptions

A1  $\|Z^\star\| = O(\sqrt{n})$, i.e. the spectral norm of $Z^\star$ is of the order of $\sqrt{n}$.

A2.  $\sigma_{r+1}(X) = O(\sqrt{n})$ with probability at least $1 - \delta$.

The MMC model makes the following assumptions. These assumptions are the same as in the main paper. We enumerate it here for the sake of clarity.

M1.  $X = M^\star + N$.

M2.  $\mathbb{E}N = 0$.

M3.  $M_{i,j}^\star = g^\star(Z_{i,j}^\star) \ \forall i = [n], j = [m]$.

M4.  Assume that $n \geq m$, and $rank(Z^\star) = r \ll m$.

M5.  Boundedness assumption: $|Z_{i,j}^\star| \leq 1, |X_{i,j}| \leq 1$ for all $i \in [n], j \in [m]$.

M6.  $g^\star : \mathbb{R} \to \mathbb{R}$ is monotonic and 1-Lipschitz.

M7. The set $\Omega$ is generated by sampling uniformly at random with replacement from the index set $[n] \times [m]$.

We would like to remark that our $1-$ Lipschitz assumption made in M6, has been done just to simplify our final expression. The proofs that we present here go though with minor changes for the general $L-$ Lipschitz case.

## 2.1 Notation

All of our matrices, unless explicitly stated, will be $n \times m$ with $n \geq m$. $||A||$ is the spectral norm of matrix $A$, and $||A||_\star$ is the nuclear norm of matrix $A$. We shall denote by $\sigma_1(A) \geq \sigma_2(A) \geq \ldots$ the singular values of matrix $A$. Finally, let $\mathcal{G} \stackrel{\text{def}}{=} \{g \mid g : [-W, W] \to [-1, 1]$ is monotonic and 1-Lipschitz$\}$.

## 2.2 Main Theorem

**Theorem 1.** *Denote by* $\mu_1 = \mathbb{E}||N||, \mu_2 = \mathbb{E}||N||^2$. *Let* $\alpha = ||M^\star - Z^\star||$. *Then, under assumptions A1, A2, and modeling assumptions M1-M7, the MSE of the estimator output by* $MMC$ *with* $T = 1$ *is given by*

$$MSE(\hat{M}) = O\Big(\sqrt{\frac{r}{m}} + \frac{\sqrt{mn\log(n)}}{|\Omega|} + \frac{mn}{|\Omega|^{3/2}} + \frac{\sqrt{mn}}{|\Omega|} + \sqrt{\frac{r}{m\sqrt{n}}\left(\mu_1 + \frac{\mu_2}{\sqrt{n}}\right)} + \sqrt{\frac{r\alpha}{m\sqrt{n}}\left(1 + \frac{\alpha}{\sqrt{n}}\right)} + \sqrt{\frac{rmn\log^2(n)}{|\Omega|^2}}\Big) \tag{5}$$

## 2.3 Interpretation of our results

In order to obtain best results, we assume that $\alpha = O(\sqrt{n})$. Typically $\mu_1 = O(\sqrt{n})$, and $\mu_2 = O(n)$. In these settings we can simplify the above expression to

$$MSE(\hat{M}) = O\left(\frac{\sqrt{mn\log(n)}}{|\Omega|} + \frac{mn}{|\Omega|^{3/2}} + \frac{\sqrt{mn}}{|\Omega|} + \sqrt{\frac{r}{m}} + \sqrt{\frac{rmn\log^2(n)}{|\Omega|^2}}\right). \tag{6}$$

This result can be converted into a sample complexity bound as follows. If we are given $|\Omega| = \max\left(\frac{n^{4/3}}{\epsilon}, \frac{n\log(n)\sqrt{r}}{\epsilon^2}\right)$, then

$$MSE(\hat{M}) \leq \sqrt{\frac{r}{m}} + \epsilon. \tag{7}$$

# 3 Towards proof of Theorem (1)

We begin with the following technical lemma that will be used in the proof.

**Lemma 1.** *Let* $\mathcal{G} = \{g|g : [-W, W] \to [-1, 1]$ *is monotonic and 1-Lipschitz*$\}$. *With probability at least* $1 - \delta$ *over the sample* $z_1, \ldots, z_n$, *the following statement is true for all* $g \in \mathcal{G}$

$$\left|\frac{1}{n}\sum(g(z_i) - y_i)^2 - \mathbb{E}(g(z) - y)^2\right| = \tilde{O}\left(\sqrt{\frac{W}{n}}\right) \tag{8}$$

*where* $\tilde{O}$ *hides logarithmic dependence on* $n, W, 1/\delta$.

*Proof.* Let $\hat{\mathcal{R}}_n(\mathcal{G})$ be the empirical Rademacher complexity of function class $\mathcal{G}$, and let $\mathcal{N}_\infty(\epsilon, \mathcal{G})$ be the $L_\infty$ covering number of the function clas $\mathcal{G}$. From [1, Lemma 6] we know that

$$\mathcal{N}_\infty(\epsilon, \mathcal{G}) \leq \frac{1}{\epsilon}2^{\frac{2W}{\epsilon}}. \tag{9}$$

The above covering number allows us to bound the empirical Rademacher complexity of the function class $\mathcal{G}$ via Dudley's entropy bound. Using [2, Lemma A.3], and the fact that $\mathcal{N}_\infty(\epsilon, \mathcal{G}) \geq \mathcal{N}_2(\epsilon, \mathcal{G}, z_1, \ldots, z_n)$ we get

$$\hat{\mathcal{R}}_n(\mathcal{G}) \leq \inf_{\alpha \geq 0} 4\alpha + 10 \int_\alpha^1 \sqrt{\frac{\log \mathcal{N}_\infty(\epsilon, \mathcal{G})}{n}} \, d\epsilon \tag{10}$$

$$\leq 4\alpha + 10 \int_\alpha^1 \sqrt{\frac{\frac{2W}{\epsilon} \log(\frac{1}{\epsilon})}{n}} \, d\epsilon \tag{11}$$

$$\leq 4\alpha + 10 \sqrt{\frac{2W}{n}} \int_\alpha^1 \frac{1}{\epsilon} \, d\epsilon \tag{12}$$

$$\leq 10 \sqrt{\frac{2W}{n}} \log\left(\frac{4e}{10} \sqrt{\frac{n}{2W}}\right). \tag{13}$$

Using a uniform convergence bound in terms of the Rademacher complexity of the function class [3, Theorem 8] we get the desired result. $\qquad\square$

**Lemma 2.** *Let* $\epsilon_2 = \mathbb{E}[\frac{1}{mn} \sum_{i,j} (\hat{Z}_{i,j} - Z_{i,j}^\star)^2]$. *Then, under assumptions A1-A2 and M1-M8, we have*

$$MSE(\hat{M}) \leq O\left(\frac{\sqrt{mn \log(n)}}{|\Omega|} + \sqrt{\frac{n}{|\Omega|}} + \frac{mn}{|\Omega|^{3/2}} + \frac{\sqrt{mn}}{|\Omega|} + \epsilon_2 + \sqrt{\epsilon_2}\right)$$

*Proof.*

$$\frac{1}{mn} \mathbb{E}\left[\sum_{i,j} (\hat{M}_{i,j} - M_{i,j}^\star)^2\right] = \frac{1}{mn} \mathbb{E}\left[\sum_{i,j} (\hat{g}(\hat{Z}_{i,j}) - g^\star(Z_{i,j}^\star))^2\right]$$

$$= \frac{1}{mn} \mathbb{E}\left[\sum_{i,j} \left(\hat{g}(\hat{Z}_{i,j}) - g^\star(\hat{Z}_{i,j}) + g^\star(\hat{Z}_{i,j}) - g^\star(Z_{i,j}^\star)\right)^2\right]$$

$$\leq 2\,\mathbb{E}\underbrace{\left[\frac{1}{mn} \sum_{i,j} \left(\hat{g}(\hat{Z}_{i,j}) - g^\star(\hat{Z}_{i,j})\right)^2\right]}_{T_1} + 2\,\mathbb{E}\underbrace{\left[\frac{1}{mn} \left(g^\star(\hat{Z}_{i,j}) - g^\star(Z_{i,j}^\star)\right)^2\right]}_{T_2}$$

$$= 2T_1 + 2T_2.$$

We shall bound $T_2$ in terms of $\epsilon_2$.

**Bounding** $T_2$**:**

$$T_2 = \frac{1}{mn} \mathbb{E} \sum_{i,j} (g^\star(\hat{Z}_{i,j}) - g^\star(Z_{i,j}^\star))^2 \tag{14}$$

$$\overset{(a)}{\leq} \frac{1}{mn} \mathbb{E} \sum_{i,j} (\hat{Z}_{i,j} - Z_{i,j}^\star)^2 \overset{\text{def}}{=} \epsilon_2 \tag{15}$$

where inequality (a) follows from the fact that $g^\star$ is 1-Lipschitz. Next we shall bound $T_1$ in terms of $\epsilon_2$ and other terms.

**Bounding $T_1$:**

$$\mathbb{E}\left[\frac{1}{mn}\sum_{i,j}\left(\hat{g}(\hat{Z}_{i,j})-g^{\star}(\hat{Z}_{i,j})\right)^2\right]=\underbrace{\mathbb{E}\left[\frac{1}{|\Omega|}\sum_{\Omega}\left(\hat{g}(\hat{Z}_{i,j})-g^{\star}(\hat{Z}_{i,j})\right)^2\right]}_{T_{1,1}}+$$

$$\underbrace{\mathbb{E}\left[\frac{1}{mn}\sum_{i,j}\left(\hat{g}(\hat{Z}_{i,j})-g^{\star}(\hat{Z}_{i,j})\right)^2\right]-\mathbb{E}\left[\frac{1}{|\Omega|}\sum_{\Omega}\left(\hat{g}(\hat{Z}_{i,j})-g^{\star}(\hat{Z}_{i,j})\right)^2\right]}_{\Delta_1}$$

(16)

Next we shall bound $T_{1,1}$ as follows. Since, $\hat{g}, g^{\star}$, by definition belong to $\mathcal{G}$, and because $\hat{g}$ is the solution to the optimization problem

$$\hat{g}=\arg\min_{g\in\mathcal{G}}\sum_{\Omega}(g(\hat{Z}_{i,j})-X_{i,j})^2,$$

(17)

hence via the generalized Pythagorean inequality [4] we have

$$\sum_{\Omega}(\hat{g}(\hat{Z}_{i,j})-X_{i,j})^2+\sum_{\Omega}(\hat{g}(\hat{Z}_{i,j})-g^{\star}(\hat{Z}_{i,j}))^2\le\sum_{\Omega}(X_{i,j}-g^{\star}(\hat{Z}_{i,j}))^2.$$

(18)

Using Equation (18) we can bound $T_{1,1}$ as follows

$$T_{1,1}=\mathbb{E}\left[\frac{1}{|\Omega|}\sum_{\Omega}\left(\hat{g}(\hat{Z}_{i,j})-g^{\star}(\hat{Z}_{i,j})\right)^2\right]$$

$$\le\mathbb{E}\left[\frac{1}{|\Omega|}\sum_{\Omega}(X_{i,j}-g^{\star}(\hat{Z}_{i,j}))^2-\frac{1}{|\Omega|}\sum_{\Omega}\left(X_{i,j}-\hat{g}(\hat{Z}_{i,j})\right)^2\right]$$

$$=\underbrace{\mathbb{E}\left[\frac{1}{|\Omega|}\sum_{\Omega}(X_{i,j}-g^{\star}(\hat{Z}_{i,j}))^2\right]-\mathbb{E}\left[\frac{1}{|\Omega|}\sum_{\Omega}(X_{i,j}-g^{\star}(Z^{\star}_{i,j}))^2\right]}_{I_1}+$$

$$\underbrace{\mathbb{E}\left[\frac{1}{mn}\sum_{i,j}(X_{i,j}-g^{\star}(Z^{\star}_{i,j}))^2\right]-\mathbb{E}\left[\frac{1}{mn}\sum_{i,j}(X_{i,j}-\hat{g}(\hat{Z}_{i,j}))^2\right]}_{I_2}+$$

(19)

$$\underbrace{\mathbb{E}\left[\frac{1}{|\Omega|}\sum_{\Omega}(X_{i,j}-g^{\star}(Z^{\star}_{i,j}))^2\right]-\mathbb{E}\left[\frac{1}{mn}\sum_{i,j}(X_{i,j}-g^{\star}(Z^{\star}_{i,j}))^2\right]}_{I_3}+$$

$$\underbrace{\mathbb{E}\left[\frac{1}{mn}\sum_{i,j}(X_{i,j}-\hat{g}(\hat{Z}_{i,j}))^2\right]-\mathbb{E}\left[\frac{1}{|\Omega|}\sum_{\Omega}(X_{i,j}-\hat{g}(\hat{Z}_{i,j}))^2\right]}_{I_4}$$

We shall look at the terms $I_1, I_2, I_3, I_4$ and bound them separately. From assumption A1 we know that $g^{\star}(Z^{\star}_{i,j})$ is the best estimator of $X_{i,j}$ in mean squared. Hence, $I_2\le 0$. We next bound $I_1, I_3, I_4$. $|Z^{\star}|_{\infty}\le 1$, and $|X^{\star}|_{\infty}\le 1$, hence $|X_{i,j}-g^{\star}(Z^{\star}_{i,j})|\le 2$. If we call $\Delta_3$ the random variable whose expectation is $I_3$, then $\Delta_3\le 4$ surely. Moreover we can apply lemma (1) to guarantee that $\Delta_3\le O\left(\sqrt{\frac{\log(|\Omega|/\delta)}{|\Omega|}}\right)$ with probability at least $1-\delta$. Choose $\delta=\frac{1}{\sqrt{|\Omega|}}$. We then have

$$I_3=\mathbb{E}\Delta_3\le 4\delta+(1-\delta)O\left(\sqrt{\frac{\log(|\Omega|/\delta)}{|\Omega|}}\right)=O\left(\sqrt{\frac{\log(|\Omega|)}{|\Omega|}}\right).$$

(20)

Next, we bound $I_4$. This needs a slightly careful treatment, since $\hat{Z}_{i,j}$ is random. Let $A = \frac{1}{p|\Omega|} X \circ \Delta$. Let $A = \sum \sigma_i u_i v_i^\top$ be the SVD of $A$ with $\sigma_1 \geq \sigma_2 \geq \cdots \sigma_m$. By definition $\hat{Z} = P_r(A)$. Hence, $A - Z = \sum_{i \geq r+1} \sigma_i u_i v_i^\top$. This means that

$$
\begin{aligned}
|A - \hat{Z}|_\infty &\leq ||A - \hat{Z}|| \\
&= || \sum_{i \geq r+1} \sigma_i u_i v_i^\top || \\
&= \sigma_{r+1} \\
&\leq \sigma_1(A - X) + \sigma_{r+1}(X),
\end{aligned}
\tag{21}
$$

where we used Weyl's inequality to get the last line from the penultimate line. We shall now use the above bound on $\hat{Z} - A$ to obtain upper bound on $|\hat{Z}|_\infty$ as follows

$$
\begin{aligned}
|\hat{Z}|_\infty &\overset{(a)}{\leq} |\hat{Z} - A|_\infty + |A - X|_\infty + |X|_\infty \\
&\overset{(b)}{\leq} ||A - X|| + |X|_\infty + ||A - X|| + \sigma_{r+1}(X) \\
&= 2||A - X|| + |X|_\infty + \sigma_{r+1}(X) \\
&= 2||A - X|| + 1 + \sigma_{r+1}(X) \\
&\overset{(c)}{\leq} 2||A - X|| + 1 + \sigma_{r+1}(X)
\end{aligned}
\tag{22}
$$

To obtain inequality (a) we used the triangle inequality, and to obtain inequality (b) we used Equation (21). Now, consider the event

$$
\mathcal{E}_1 = \left\{ ||A - X|| \leq \frac{2mn \log\left(\frac{m+n}{\delta}\right)}{3|\Omega|} + \sqrt{\frac{2 \log(\frac{m+n}{\delta})mn}{|\Omega|}} \right\}.
\tag{23}
$$

From Lemma 5 we know that conditioned on $X$, $\mathbb{P}(\mathcal{E}_1) \geq 1 - \delta$ over the randomness in $\Omega$. Using equation (22) we get that on event $\mathcal{E}_1$

$$
|\hat{Z}|_\infty = O\left( \sigma_{r+1}(X) + \frac{mn \log(\frac{m+n}{\delta})}{|\Omega|} + \sqrt{\frac{mn \log((m+n)/\delta)}{3|\Omega|}} \right) \overset{\text{def}}{=} b
\tag{24}
$$

Now let $I_4'$ be the argument to the expectation operator in $I_4$. Let us define another event

$$
\mathcal{E}_{11} = \left\{ I_4' \leq \sqrt{\frac{b \log((m+n)/\delta)}{|\Omega|}} \right\}
\tag{25}
$$

By arguments similar to the one used in lemma (1), we get that $\mathbb{P}(\mathcal{E}_{11}) \geq 1 - \delta$ over the random choice of $\Omega$. Notice that $I_4' \leq 4$ surely. We are now ready to calculate $I_4$ as follows

$$
\begin{aligned}
I_4 &= \mathbb{E}_X \mathbb{E}_{\Omega|X} I_4' \\
&\leq \mathbb{E}_X \mathbb{P}(\mathcal{E}_1) \mathbb{E}_{\Omega|X, \mathcal{E}_1} I_4' + 4\mathbb{P}(\bar{\mathcal{E}}_1) \\
&\leq \mathbb{E}_X \mathbb{P}(\mathcal{E}_1)(\mathbb{P}(\mathcal{E}_{11}) I_4' + 4\mathbb{P}(\bar{\mathcal{E}}_{11})) + 4\mathbb{P}(\bar{\mathcal{E}}_1) \\
&\leq 8\delta + \mathbb{E}_X \sqrt{\frac{b \log((m+n)/\delta)}{|\Omega|}}
\end{aligned}
\tag{26}
$$

Substituting the value of $b$, and using $\delta = \frac{1}{|\Omega|}$, and using assumption A2, we get that

$$
I_4 = \mathbb{E}_X \mathbb{E}_{\Omega|X} I_4'
\tag{27}
$$

$$
\leq 8\delta + \mathbb{E}_X \sqrt{\frac{1}{|\Omega|} O\left( \sigma_{r+1}(X) + \frac{mn \log((m+n)|\Omega|)}{|\Omega|} + \sqrt{\frac{mn \log((m+n)|\Omega|)}{3|\Omega|}} \right)}
\tag{28}
$$

$$
= O\left( \sqrt{\frac{mn}{|\Omega|^2}} \log^2((m+n)|\Omega|) \right)
\tag{29}
$$

Notice that $\Delta_1$ uses $\hat{g} - g^\star$ which is a 2 Lipchitz function. By perfoming a similar analysis as in $I_4$ it is easy to show that $\Delta_1 = O(I_4)$.

**Bounding $I_1$.**

$$I_1 = \mathbb{E}\left[\frac{1}{|\Omega|}\sum_\Omega (X_{i,j} - g^\star(\hat{Z}_{i,j}))^2 - \frac{1}{|\Omega|}\sum_\Omega (X_{i,j} - g^\star(Z^\star_{i,j}))^2\right] \tag{30}$$

$$= \mathbb{E}\left[\frac{1}{|\Omega|}\sum_\Omega (g^\star(Z^\star_{i,j}) - g^\star(\hat{Z}_{i,j}))(2X_{i,j} - g^\star(\hat{Z}_{i,j}) - g^\star(Z^\star_{i,j}))\right] \tag{31}$$

$$\overset{(a)}{\leq} 4\mathbb{E}\frac{1}{|\Omega|}|g^\star(Z^\star_{i,j}) - g^\star(\hat{Z}_{i,j})| \tag{32}$$

$$\overset{(b)}{\leq} 4\mathbb{E}\frac{1}{|\Omega|}\sum_\Omega |Z^\star_{i,j} - \hat{Z}_{i,j}| \tag{33}$$

$$= 4\mathbb{E}\frac{1}{mn}\sum_{i,j}|Z^\star_{i,j} - \hat{Z}_{i,j}| + 4\underbrace{\left(\mathbb{E}\frac{1}{|\Omega|}\sum_\Omega |Z^\star_{i,j} - \hat{Z}_{i,j}| - \mathbb{E}\frac{1}{mn}\sum_{i,j}|Z^\star_{i,j} - \hat{Z}_{i,j}|\right)}_{\Delta_5} \tag{34}$$

$$\overset{(c)}{\leq} 4\mathbb{E}\frac{1}{mn}\sum_{i,j}|Z^\star_{i,j} - \hat{Z}_{i,j}| + 4\Delta_5 \tag{35}$$

$$\overset{(d)}{\leq} 4\sqrt{\mathbb{E}\frac{1}{mn}\sum_{i,j}|Z^\star_{i,j} - \hat{Z}_{i,j}|^2} + 4\Delta_5 = 4(\sqrt{\epsilon_2} + \Delta_5) \tag{36}$$

where, to get inequality (a) we used the fact that $|X_{i,j}| \leq 1$ and $|g^\star| \leq 1$. To get inequality (b) we used the fact that $g^\star$ is 1 Lipschitz. To get inequality (c) we used concentration of measure. Finally, to get inequality (d) we used Jensen's inequality to bound $\mathbb{E}|x| \leq \sqrt{Ex^2}$. Our next step is to bound $\Delta_5$.

**Bounding $\Delta_5$:** The idea is to consider the event $\mathcal{E}_1$ as was done during bounding the term $I_4$. Once again we shall consider the event

$$\mathcal{E}_1 = \left\{||A - X|| \leq \frac{2mn\log\left(\frac{m+n}{\delta}\right)}{3|\Omega|} + \sqrt{\frac{2\log(\frac{m+n}{\delta})mn}{|\Omega|}}\right\}. \tag{37}$$

Similar to arguments there, we know from Equation (24) that on event $\mathcal{E}_1$

$$|\hat{Z}|_\infty = O\left(\sigma_{r+1}(X) + \frac{mn\log(\frac{m+n}{|\Omega|})}{|\Omega|} + \sqrt{\frac{mn\log((m+n)/\delta)}{3|\Omega|}}\right) \overset{\text{def}}{=} b$$

Consider the collection of random variables $\xi_1, \ldots, \xi_{|\Omega|}$, where each $\xi_k$ takes the value $Z^\star_{i,j} - \hat{Z}_{i,j}$, where $(i,j)$ is chosen u.a.r. with replacement from $[n] \times [m]$. It is easy to see that each of $\xi_k \in [0, b+1]$ on $\mathcal{E}_1$. Applying Hoeffding inequality we get on $\mathcal{E}_1$ with probability at least $1 - \delta$ over the random choice of $\Omega$, and on event $\mathcal{E}_1$

$$\frac{1}{|\Omega|}\sum_\Omega |Z^\star_{i,j} - \hat{Z}_{i,j}| - \sum_{i,j}|Z^\star_{i,j} - \hat{Z}_{i,j}| \leq \sqrt{\frac{(b+1)^2}{2|\Omega|}\log(1/\delta)} \tag{38}$$

By arguments similar to the ones used in establishing bounds for $I_4$, we get

$$\Delta_5 \leq O\left(\log\left((m+n)|\Omega|\right)\left(\sqrt{\frac{n}{|\Omega|}} + \frac{mn}{|\Omega|^{3/2}} + \frac{\sqrt{mn}}{|\Omega|}\right)\right). \tag{39}$$

This concludes our first set of calculations. With this we have

$$MSE(\hat{M}) = O\left(\frac{\sqrt{mn\log(n)}}{|\Omega|} + \sqrt{\frac{n}{|\Omega|}} + \frac{mn}{|\Omega|^{3/2}} + \frac{\sqrt{mn}}{|\Omega|} + \epsilon_2 + \sqrt{\epsilon_2}\right) \tag{40}$$

$\square$

The rest of the proof establishes upper bounds on $\epsilon_2$.

### 3.1 Bounding $\epsilon_2$.

In order to establish an upper bound on $\epsilon_2$ we first need the following projection lemma. This lemma is similar in spirit to a lemma of S.Chatterjee [5, Lemma 3.5]. Before we establish this lemma, we would like to clarify the notation that we use. Let $\sigma_1 \geq \sigma_2 \geq \sigma_m$ be the singular values of a matrix $A$.

**Lemma 3.** *Let $A = \sum_{i=1}^{m} \sigma_i x_i y_i^\top$ be the SVD of a known, rectangular matrix $A \in \mathbb{R}^{\times m}$, with the singular values $\sigma_1 \geq \sigma_2 \ldots \geq \sigma_m$ arranged in decreasing order. Let $B$ be an unknown $n \times m$ matrix. Given $1 \leq r \leq m$, let $\hat{B} \stackrel{\text{def}}{=} P_r(A) \stackrel{\text{def}}{=} \sum_{i=1}^{r} \sigma_i x_i y_i^\top$ be the projection estimator of $B$. Then,*

$$||P_r(A) - B||_F \leq \sqrt{||B||_\star(\sigma_{r+1} + ||A - B||)} + 2\sqrt{2r}(\sigma_{r+1} + ||A - B||). \tag{41}$$

*Proof.* Let $B = \sum_{i=1}^{m} \tau_i u_i v_i^\top$ be the SVD of $B$ with $\tau_1 \geq \tau_2 \geq \ldots \tau_m$. Let $G = P_r(B) \stackrel{\text{def}}{=} \sum_{i=1}^{r} \tau_i u_i v_i^\top$ be the projection of matrix $B$ onto the rank $r$ cone.

$$||\hat{B} - B||_F \leq ||\hat{B} - G||_F + ||G - B||_F, \tag{42}$$

and

$$||G - B||_F^2 = ||\sum_{i \geq r+1} \tau_i u_i v_i^\top||_F^2 = \sum_{i \geq r+1} \tau_i^2 \leq (\max_{i \geq r+1} \tau_i)||B||_\star. \tag{43}$$

Let $\delta_1 \geq \delta_2 \geq \ldots$ be the singular values of $A - B$ in decreasing order. Then from Weyl's inequality we know that

$$\max_i |\sigma_i - \tau_i| \leq \max_i \delta_i = ||A - B||. \tag{44}$$

Hence, for $i \geq r + 1$,

$$\tau_i \leq \sigma_i + ||A - B|| \leq \sigma_{r+1}(A) + ||A - B||. \tag{45}$$

This allows us to conclude that $\max_{i \geq r+1} \tau_i \leq \sigma_{r+1}(A) + ||A - B||$. Combined with Equation (43) we get

$$||G - B||_F^2 \leq ||B||_\star(\sigma_{r+1}(A) + ||A - B||). \tag{46}$$

Next, we shall upper bound the quantity $||\hat{B} - G||_F$. By construction, both $\hat{B}$ and $G$ are rank $r$ matrices and hence $\hat{B} - G$ has rank at most $2r$ matrix. This allows us to control the Frobenius norm of $\hat{B} - G$ via its spectral norm as follows

$$||\hat{B} - G||_F \leq \sqrt{2r}||\hat{B} - G|| \tag{47}$$

To bound $||\hat{B} - G||$ consider the following decomposition

$$||\hat{B} - G|| \leq ||\hat{B} - A|| + ||A - B|| + ||B - G||. \tag{48}$$

We have

$$||\hat{B} - A|| = ||\sum_i \sigma_i x_i y_i^\top|| \leq \sigma_{r+1}. \tag{49}$$

$$||B - G|| = ||\sum_{i \geq r+1} \tau_i u_i v_i^\top|| = \tau_{r+1} \stackrel{\text{(a)}}{\leq} \sigma_{r+1} + ||A - B|| \tag{50}$$

where to get inequality (a) we used Equation (45). Combining Equations (48), (49), (50) we get

$$||\hat{B} - G|| \leq \sigma_{r+1} + ||A - B|| + \sigma_{r+1} + ||A - B|| = 2(\sigma_{r+1} + ||A - B||) \tag{51}$$

and using Equation (47) we get

$$||\hat{B} - G||_F \leq 2\sqrt{2r}(\sigma_{r+1} + ||A - B||) \tag{52}$$

Finally using Equation (46) and Equation (52) we get

$$||\hat{B} - B||_F \leq 2\sqrt{2r}(\sigma_{r+1} + ||A - B||) + \sqrt{||B||_\star(\sigma_{r+1} + ||A - B||)}. \tag{53}$$

$\square$

Let us define matrices $A, B, \hat{Z}$ as follows

$$A \overset{\text{def}}{=} \frac{1}{p|\Omega|} X \circ \Delta_\Omega \tag{54}$$

$$B \overset{\text{def}}{=} Z^\star \tag{55}$$

$$\hat{Z} \overset{\text{def}}{=} P_r(A). \tag{56}$$

In the rest of the document, unless otherwise stated, the above definitions of $A, B, \hat{Z}$ will be applicable. In order to obtain an upper bound on $\epsilon_2$ we shall use Lemma (3) with the above choices for $A, B$.

$$||\hat{Z} - Z^\star||_F \leq \sqrt{||Z^\star||_\star (\sigma_{r+1} + ||A - Z^\star||)} + 2\sqrt{2r}(\sigma_{r+1} + ||A - Z^\star||). \tag{57}$$

Since $Z^\star$ is of rank $r$, we have $||Z^\star||_\star \leq r||Z^\star||$. From triangle inequality $||A|| \leq ||A - Z^\star|| + ||Z^\star||$. These facts coupled with the fact that $\sigma_{r+1} \leq \sigma_1$ allows us to obtain

$$\mathbb{E}||\hat{Z} - Z^\star||_F^2 \leq r||Z^\star||(2\mathbb{E}||A - Z^\star|| + ||Z^\star||) + 8r(4\mathbb{E}||A - Z^\star||^2 + ||Z^\star||^2). \tag{58}$$

Notice that $\epsilon_2$ is a scaled version of $\mathbb{E}||\hat{Z} - Z^\star||_F^2$. Let,

$$\beta_1 \overset{\text{def}}{=} \mathbb{E}||A - X|| \tag{59}$$

$$\beta_2 \overset{\text{def}}{=} \mathbb{E}||A - X||^2. \tag{60}$$

Using the above definitions, Equation (58), the triangle inequality $||A - Z^\star|| \leq ||A - X|| + ||X - Z^\star||$, along with the elementary fact that $(a+b)^2 \leq 2a^2 + 2b^2$, we obtain

$$\mathbb{E}||\hat{Z} - Z^\star||_F^2 \leq r||Z^\star||(2\beta_1 + 2\mathbb{E}||X - Z^\star|| + ||Z^\star||) + 8r(8\beta_2 + 8\mathbb{E}||X - Z^\star||^2 + ||Z^\star||^2) \tag{61}$$

$$= r||Z^\star||(2\mathbb{E}||X - Z^\star|| + ||Z^\star||) + 8r(8\mathbb{E}||X - Z^\star||^2 + ||Z^\star||^2) + r(2\beta_1 + 64\beta_2). \tag{62}$$

**Bounding** $\beta_1, \beta_2$. In order to bound $\beta_1, \beta_2$ we need upper bounds on spectral norm of sums of random matrices. Towards this, the following Bernstein inequality is useful

**Theorem 2** (Bernstein's inequality). *Let $S_1, \ldots S_k$ be independent, centered random matrices with common dimension $n \times m$, and assume that each one of them is bounded*

$$||S_j|| \leq L \text{ for each } j \geq 1. \tag{63}$$

*Let $M = \sum_{j=1}^k S_j$, and let $\nu(M)$ denote the matrix variance statistic of the sum*

$$\nu(M) = \max\left\{||\sum_{j=1}^k \mathbb{E}S_j S_j^\top||, ||\sum_{j=1}^k \mathbb{E}S_j^\top S_j||\right\}. \tag{64}$$

*Then*

*1.*

$$\mathbb{P}(||M|| \geq t) \leq (m+n)\exp\left(\frac{-t^2/2}{\nu(M) + Lt/3}\right), \tag{65}$$

*Furthermore*

*2.*

$$\mathbb{E}Z \leq \sqrt{2\nu(M)\log(m+n)} + \frac{1}{3}L\log(m+n). \tag{66}$$

We shall bound $\beta_1$ using part (ii) of Bernstein's inequality, and $\beta_2$ using part (ii) of Bernstein's inequality. The next two lemma's provide necessary material for bounding $\beta_1, \beta_2$.

**Proposition 1.** *Let $\Delta$ be a random mask of size $n \times m$, where a random location is chosen and set to 1, and rest of the entries are set to 0. Let $X$ be a matrix of size $n \times m$ with entries bounded in absolute value by 1. Define $S = \frac{1}{p}X \circ \Delta - X$. Let $p = \frac{1}{mn}$. Then,*

1. $||S|| \leq ||X|| + \frac{1}{p}$

2. $\mathbb{E}S^\top S = \mathbb{E}SS^\top = X \circ X - XX^\top$

*Proof.*

$$||S|| = ||\frac{1}{p}X \circ \Delta - X|| \leq ||X|| + ||\frac{1}{p}X \circ \Delta|| \overset{(a)}{\leq} ||X|| + \frac{1}{p}. \tag{67}$$

In the above set of inequalities in order to derive (a) we used the fact that $X \circ \Delta$ is an $n \times m$ matrix with a single non-zero entry bounded in absolute value by 1. Hence the spectral norm of this matrix will be bounded by 1. To derive the second part of the proposition we proceed as follows

$$\mathbb{E}SS^\top = \mathbb{E}(\frac{1}{p}X \circ \Delta - X)(\frac{1}{p}X \circ \Delta - X)^\top = \mathbb{E}[\frac{1}{p^2}(X \circ \Delta)(X \circ \Delta)^\top - \frac{1}{p}(X \circ \Delta)X^\top - \frac{1}{p}X(X \circ \Delta)^\top + XX^\top]. \tag{68}$$

Via elementary calculations, it is easy to verify that

$$\mathbb{E}\left[\frac{1}{p^2}(X \circ \Delta)(X \circ \Delta)^\top\right] = X \circ X \tag{69}$$

$$\mathbb{E}\left[\frac{1}{p}X(X \circ \Delta)^\top\right] = \mathbb{E}\left[\frac{1}{p}(X \circ \Delta)X^\top\right] = XX^\top. \tag{70}$$

These identities allow us to conclude part (ii) of this proposition. $\square$

We are now ready to bound the quantities $\beta_1, \beta_2$

**Lemma 4.** *Let $p = \frac{1}{mn}$. Then,*

$$\beta_1 = \mathbb{E}\left\|\frac{1}{p|\Omega|}X \circ \Delta_\Omega - X\right\| \leq \sqrt{\frac{2\log(m+n)||X \circ X - XX^\top||}{|\Omega|}} + \frac{\log(m+n)(p||X|| + 1)}{3p|\Omega|}. \tag{71}$$

*Proof.*

$$\left\|\frac{1}{p|\Omega|}X \circ \Delta_\Omega - X\right\| = \frac{1}{|\Omega|}\left\|\sum_{j=1}^{|\Omega|}\underbrace{(X \circ \Delta_j - X)}_{S_j}\right\| \tag{72}$$

Here $\Delta_1, \ldots, \Delta_\Omega$ are random i.i.d. boolean masks with each of them having exactly one non-zero, whose location is chosen uniformly at random from $[n] \times [m]$. For this reason the matrices $S_1, \ldots, S_{|\Omega|}$ are i.i.d. matrices. It is easy to see that $\mathbb{E}S_j = 0$ for each $j \geq 1$. Applying Bernstein's inequality (Theorem (2)) and using Proposition (1) to bound the necessary quantities we get that

$$\mathbb{E}\left\|\frac{1}{p|\Omega|}X \circ \Delta_\Omega - X\right\| = \frac{1}{|\Omega|}\left[\sqrt{2\log(m+n)|\Omega|||X \circ X - XX^\top||} + \frac{\log(m+n)}{3}(||X|| + \frac{1}{p})\right] \tag{73}$$

$$= \sqrt{\frac{2\log(m+n)||X \circ X - XX^\top||}{|\Omega|}} + \frac{\log(m+n)(p||X|| + 1)}{3p|\Omega|} \tag{74}$$

$\square$

Next we bound $\beta_2$.

**Lemma 5.** *With probability at least $1 - \delta$*

$$\left\|\frac{1}{p|\Omega|}X \circ \Delta_\Omega - X\right\| \leq \frac{2\log\left(\frac{m+n}{\delta}\right)}{3|\Omega|}\left(||X|| + \frac{1}{p}\right) + \sqrt{\frac{4\log(\frac{m+n}{\delta})mn}{|\Omega|}}. \tag{75}$$

*Furthermore, conditioned on $X$,*

$$\beta_2 = \mathbb{E}\left\|\frac{1}{p|\Omega|}X \circ \Delta_\Omega - X\right\|^2 \leq 1 + \left(\frac{20mn\log(n)}{3|\Omega|}\right)^2 + \frac{10\log(n)}{|\Omega|}||X \circ X - XX^\top||. \tag{76}$$

*Proof.* The first part of the theorem follows immediately by using part (i) of Bernstein's inequality. We get that, for any $\delta > 0$, with probability at least $1 - \delta$,

$$\|A - X\| \leq \frac{2\log\left(\frac{m+n}{\delta}\right)}{3|\Omega|}\left(\|X\| + \frac{1}{p}\right) + \sqrt{\frac{2\log(\frac{m+n}{\delta})\|X \circ X - XX^\top\|}{|\Omega|}}. \tag{77}$$

Now, using the fact that $\|X \circ X\| \leq \|X\|^2$ and the fact that each element of $X$ is bounded by 1 in absolute value, it follows that $\|X \circ X - XX^\top\| \leq 2mn$. Substituting this bound in the above expression allows us to get the first part of the theorem. In order to derive the second part of the theorem, we proceed as follows. We first derive a worst case upper bound on $\|A - X\|$ as follows

$$\|A - X\| = \|\frac{1}{p|\Omega|}X \circ \Delta_\Omega - X\| \tag{78}$$

$$\leq \frac{1}{p|\Omega|}\|X \circ \Delta_\Omega\| + \|X\| \tag{79}$$

$$\leq \frac{1}{p|\Omega|}\sum_{j=1}^{|\Omega|}\|X \circ \Delta_j\| + \|X\| \tag{80}$$

$$\leq \frac{1}{p} + \|X\|. \tag{81}$$

Using equations (77) and (78) we get

$$\mathbb{E}\|A - X\|^2 \leq (1 - \delta)\left(\frac{2\log\left(\frac{m+n}{\delta}\right)}{3|\Omega|}\left(\|X\| + \frac{1}{p}\right) + \sqrt{\frac{2\log(\frac{m+n}{\delta})\|X \circ X - XX^\top\|}{|\Omega|}}\right)^2 + \delta\left(\frac{1}{p} + \|X\|\right)^2 \tag{82}$$

Since each element of $X$ is bounded by 1 in magnitude, we get that $\|X\| \leq \sqrt{mn}$. Now, replace $p = \frac{1}{mn}$ and choose $\delta = \frac{1}{\left(mn + \sqrt{mn}\right)^2}$. Using the inequality $(a + b)^2 < 2a^2 + 2b^2$ and over-approximating we get the desired result. $\qquad\square$

**Final bound on $\epsilon_2$.** We are now ready to establish a bound on $\epsilon_2$. In the next bound we shall no longer keep track of explicit constants. Instead in the following calculations we shall use a universal constant $C > 0$ whose value can change from one line to another.

**Lemma 6.** *Let* $\mu_1 = \mathbb{E}\|N\|, \mu_2 = \mathbb{E}\|N\|^2$. *Then, for some universal constant $C > 0$ we have*

$$\epsilon_2 \leq O\left(\frac{r}{m\sqrt{n}}(\|M^\star - Z^\star\| + \mu_1) + \frac{r\|M^\star - Z^\star\|^2}{mn} + \frac{r\mu_2}{mn} + \frac{r}{m} + \frac{rmn\log^2(n)}{|\Omega|^2}\right) \tag{83}$$

*Proof.* From Equation (61) we have

$$\epsilon_2 \leq r\|Z^\star\|(2\mathbb{E}\|X - Z^\star\| + \|Z^\star\|) + 8r(8\mathbb{E}\|X - Z^\star\|^2 + \|Z^\star\|^2) + r(2\beta_1 + 64\beta_2). \tag{84}$$

Now, using Lemma (4) and (5) to bound $\beta_1, \beta_2$, we get

$$\epsilon_2 \leq \frac{Cr}{mn}\mathbb{E}\Big[\|Z^\star\|\,\|X - Z^\star\| + \|X - Z^\star\|^2 + \|Z^\star\|^2 + \sqrt{\frac{\log(n)\|X \circ X - XX^\top\|}{|\Omega|}} +$$

$$\frac{\log(n)}{3|\Omega|}(\|X\| + mn) + 1 + \frac{m^2n^2\log^2(n)}{|\Omega|^2} + \frac{\log(n)}{|\Omega|}\|X \circ X - XX^\top\|\Big]. \tag{85}$$

In the above expectation the expectation is being taken w.r.t. the randomness in $X$ due to additive noise of our model. We shall now compute the remaining expectations. For notational convenience, define $\mu_1 = \mathbb{E}\|N\|$, and $\mu_2 = \mathbb{E}\|N\|^2$. Using the fact that $X = M^\star + N$, we get

$$\mathbb{E}\|X \circ X - XX^\top\| \leq \mathbb{E}\|X \circ X\| + \mathbb{E}\|XX^\top\| \tag{86}$$

$$\overset{(a)}{\leq} \mathbb{E}\|X\|^2 + \mathbb{E}\|(M^\star + N)(M^\star + N)^\top\| \tag{87}$$

$$\overset{(b)}{\leq} \mathbb{E}[\|M^\star\|^2 + \|N\|^2 + 2\|M^\star\|\,\|N\| + \|M^\star(M^\star)^\top\| + M^\star N^\top + N(M^\star)^\top + NN^\top] \tag{88}$$

$$= 2\|M^\star\|^2 + 2\mu_2 + 4\|M^\star\|\mu_1 \tag{89}$$

where to obtain inequality (a) we used the fact that $||A \circ B|| \leq ||A||||B||$ [6, Problem 1.6.13, page 23]. To obtain inequality (b) we used sub-additivity of norms, and the fact that spectral norm is sub-multiplicative. By Jensen's inequality we get

$$\mathbb{E}\sqrt{||X \circ X - XX^\top||} \leq \sqrt{\mathbb{E}||X \circ X - XX^\top||} \leq \sqrt{2||M^\star||^2 + 2\mu_2 + 4\mu_1||M^\star||} \tag{90}$$

Finally using the sub-additivity of norms we get that

$$\mathbb{E}||X - Z^\star||^2 = \mathbb{E}||M^\star + N - Z^\star||^2 \leq 2||M^\star - Z^\star||^2 + 2\mathbb{E}||N||^2 = 2||M^\star - Z^\star||^2 + 2\mu_2 \tag{91}$$

$$\mathbb{E}||X - Z^\star|| = \mathbb{E}||M^\star + N - Z^\star|| \leq \mathbb{E}||M^\star - Z^\star|| + \mathbb{E}||N|| = \mathbb{E}||M^\star - Z^\star|| + \mu_1 \tag{92}$$

Now, putting together Equations (85), (86), (90), (91), and substituting the worst case bound $||M^\star|| = C\sqrt{mn}$, we get

$$\epsilon_2 \leq C\Big[\frac{r}{mn}||Z^\star||\left(||M^\star - Z^\star|| + \mu_1\right) + \frac{r}{mn}||M^\star - Z^\star||^2 + \frac{r}{mn}(\mu_2 + ||Z^\star||^2) +$$
$$\frac{r}{mn}\sqrt{\frac{\log(n)}{|\Omega|}}\left(mn + \mu_1\sqrt{mn} + \mu_2\right) + \frac{rmn\log^2(n)}{|\Omega|^2} + \frac{r\log(n)}{mn|\Omega|}(mn + \mu_2 + \mu_1\sqrt{mn})\Big]. \tag{93}$$

We can further simplify the above expression, by noting that the entries of $N$ are bounded by 1, and hence $\mu_1 = O(\sqrt{mn}), \mu_2 = O(mn)$. Note that in reality $\mu_1, \mu_2$ are much smaller, and one could lose a lot of information by considering their worst case values. However, in order to simplify the above bound for $\epsilon_2$ and make it interpretable, we shall selectively replace $\mu_1, \mu_2$ by $\sqrt{mn}, mn$ respectively, This allows us to gauge which terms are lower order terms and drop them. This gets us

$$\epsilon_2 \leq O\left(\frac{r}{m\sqrt{n}}(||M^\star - Z^\star|| + \mu_1) + \frac{r||M^\star - Z^\star||^2}{mn} + \frac{r\mu_2}{mn} + \frac{r}{m} + \frac{rmn\log^2(n)}{|\Omega|^2}\right) \tag{94}$$

$\square$

## 4 Proof of Theorem (1)

From Lemma (2) we have

$$MSE(\hat{M}) \leq O\left(\frac{\sqrt{mn\log(n)}}{|\Omega|} + \sqrt{\frac{n}{|\Omega|}} + \frac{mn}{|\Omega|^{3/2}} + \frac{\sqrt{mn}}{|\Omega|} + \epsilon_2 + \sqrt{\epsilon_2}\right)$$

From Lemma (6) we have

$$\epsilon_2 \leq O\left(\frac{r}{m\sqrt{n}}(||M^\star - Z^\star|| + \mu_1) + \frac{r||M^\star - Z^\star||^2}{mn} + \frac{r\mu_2}{mn} + \frac{r}{m} + \frac{rmn\log^2(n)}{|\Omega|^2}\right) \tag{95}$$

Putting the above two equations together we get

$$MSE(\hat{M}) = O\Big(\sqrt{\frac{r}{m}} + \frac{\sqrt{mn\log(n)}}{|\Omega|} + \frac{mn}{|\Omega|^{3/2}} + \frac{\sqrt{mn}}{|\Omega|} + \sqrt{\frac{r}{m\sqrt{n}}\left(\mu_1 + \frac{\mu_2}{\sqrt{n}}\right)} +$$
$$\sqrt{\frac{r\alpha}{m\sqrt{n}}\left(1 + \frac{\alpha}{\sqrt{n}}\right)} + \sqrt{\frac{rmn\log^2(n)}{|\Omega|^2}}\Big) \tag{96}$$

## 5 Source for datasets

Here is where one can download the real world datasets on which all of our experiments were performed.

1. Paper recommendation dataset:http://www.comp.nus.edu.sg/~sugiyama/SchPaperRecData.html.
2. Jester dataset: http://goldberg.berkeley.edu/jester-data/.
3. Movie lens dataset: http://grouplens.org/datasets/movielens/
4. Cameraman dataset: http://www.utdallas.edu/~cxc123730/mh_bcs_spl.html

Figure 1: The RMSE of $MMC - c$ when the probability of sampling each entry in the matrix is $p = 0.35$. Here we show the decay of RMSE for different values of $c$, and when the transfer function is $g^\star(Z_{i,j}) = \frac{1}{1+\exp(-cZ_{i,j})}$.

## 6  RMSE of $MMC - c$ with iterations

The RMSE of $MMC - c$ shows a decreasing trend with the number of iterations. For mild non-linearities, we in fact see linear decay of RMSE, as can be seen in Figure 1