[Reviews · NeurIPS 2015]

Submitted by Assigned_Reviewer_1

This paper proposed extends the single index model in statistics to low-rank matrix estimation. Particularly, the authors proposed to recover the underlying low-rank matrix given the observations after nonlinear transformation. An algorithm based on alternative minimization is proposed to estimate both the transformation function and the underlying low-rank matrix. Theoretical analysis is given for the proposed algorithm. Finally, experiments on both synthetic data and benchmark datasets are conducted to demonstrate the effectiveness of the proposed algorithm in recovering the matrix.

Major comments

1. A quick tip is that the bound in (18) can be greatly simplified. For example, the forth term is dominated by the second term in the setting of this paper.

2. Based on the proof in the appendix, the error bound obtained in Theorem 1 should hold with high probability. The authors should point this out explicitly.

3. As the authors also mention in the paper that the problem being solved is a non-convex problem, a natural question is whether the proposed algorithm will converge to a stationary point. However, this question is not answered in this paper.

4. Only the error bound of the one-step estimator ($T = 1$) is given. However, the error bound of the proposed algorithm is much more important and essential when T is larger than one. Also, as indicated by the experimental part, the one-step estimator is worse than the SVD estimator. In this sense, the analysis is not complete.

5. Line 333, the statement of "For such matrices $\mu_1 = O(), \mu_2 = O()$" is NOT obvious to me.

Further justifications are required for rigorous analysis.

6. Line 347, the statement of "If we are given ...." contradicts with the definition of $\tilde{O}$ in Line 306 that there should be a logarithmic term, let alone that a constant is missing.

7. In Line 085, the setup of the problem is that the entries in the matrix are observed with noise; however, in the experimental part, the settings correspond to noiseless observations. The authors should at least add more experiments corresponding to the settings being studied in this paper.

8. The writing and organization of the proof (Appendix) in this paper should be improved significantly. Based on the current writing, it is almost impossible to check the correctness of the theoretical analysis.

i) Line 104, it is not clear what A1-A8 refer to.

ii) Line 226 (Appendix), it is not evident what I'_4 is, which leads to confusion over statement in line 233.

iii) Line 191, "Hence, A - Z" ---> "Hence, A - \hat{Z}".

iv) (31) is redundant.

Summary: The theoretical analysis presented in this paper is not satisfying; and the numerical experiments do not serve as corroboration of the theory.

Submitted by Assigned_Reviewer_2

The paper addresses the problem of matrix completion, with observed entries being perturbed by an unknown nonlinear transformation. Assuming that such a nonlinear transformation on observed entries are Lipschitz, the paper proposes a method that alternates between finding the nonlinear transformation and low-rank matrix completion. The paper uses an ADMM optimization framework to implement the framework and demonstrates the effectiveness of the approach by experiments on synthetic and real data. The reviewer believes that the paper is well-written and well-organized. The idea of the paper is sufficiently novel, the approach is interesting and results show improvement with respect to the state of the art.
Summary: The paper addresses the problem of nonlinear matrix completion. The reviewer believes that the paper is well-written and well-organized. The idea of the paper is sufficiently novel, the approach is interesting and results show improvement with respect to the state of the art.

Submitted by Assigned_Reviewer_3

The paper is very well-written and easy to follow. The ideas are explained in crisp and concise manner.

The MSE analysis of section 4 is restricted to T = 1, but in practice T > 1 is the interesting case as shown by Table 1. How does the analysis extend to T > 1?

One argument made in favour of the new loss is that it does not require the derivative of gt, which is claimed to be a good thing since it is less smooth than gt and hard to estimate. But empirically how does the approach in section 3.1 compare against the proposed approach? I was expecting to see a comparison in section 5.

In line 238-239 it is mentioned that the Lipschitz constant is known. But in practice how is it set?

How does this work compare to the approach in "Retargeted matrix factorization for collaborative filtering" by Koyejo, Acharyya, and Ghosh, RecSys'13, which also looks at learning in the setting of a monotonic transformation of a low rank matrix?
Summary: The paper proposes an algorithm for matrix completion in the setting where the observed matrix is generated by taking a low rank matrix and applying an element-wise nonlinear, monotonic transformation, plus noise. The algorithm optimizes a different loss than the squared error between the observed and predicted matrix entries, but the theoretical and empirical results show that the algorithm still performs well with respect to squared error.

Submitted by Assigned_Reviewer_4

Summary: The paper presents a matrix completion algorithm for the matrix whose entries are distorted by a monotonic nonlinear function. The author observed that the nonlinear transformation destroys

the low-rank structure

and hence render the existing matrix algorithm less useful. The proposed algorithm is based on calibrated loss function and demonstrated some performance advantage on some numerical examples. The author also established an error bound for the case T = 1.

Originality and significance: The idea of calibrated loss function and estimating function g from data

is borrowed from the ICML-14 paper of Agarwal et al.([22]). The novelty of this paper is a new way of estimating the function g. The proposed method seems simpler than that of Agarwal et al and is noteworthy.

The author also established an error bound for the case of T=1. However, it is the error bound showing how the error decrease as T increases that is important to establish the validity of the algorithm. Without the error bound for T > 1, it's questionable whether the algorithm will converge. Numerical example showing the error versus T is also absent in the paper. Besides,

empirical(or theoretical) study is needed to back up the following important claim: the proposed algorithm works better than simply substituting the g' using Lipschitz constant(line 170-171). Since the proposed method essentially use the Lipschitz constant to estimate g' too, it is not immediately obvious how much advantage the proposed algorithm have over the simple substitution. The author should elaborate this point in more detail.

Clarity: Overall, the paper is clear except some typos: Line 297: 'i = [ n ]' -> 'i \in [ n ]

Summary: The proposed algorithm is noteworthy but not good enough for NIPS.

Major work(both theoretical and empirical study) needs to be done to show the proposed algorithm has advantages over existing algorithms.

Author Feedback
Author rebuttal: We would like to thank all the reviewers for taking the time to provide their feedback. We first address a common concern raised by the reviewers.

Bounds for T>1: In the paper we established rigorous error bounds for the T=1 case. We would like to mention that one can simply use a validation set to keep track of the best iterate in algorithm 1, and then the bound presented in the paper (Theorem 1) also applies to all the iterates T>=1. We will clarify this in our final version. Obtaining sharper bounds than the ones presented in this paper remains a challenging open problem.

Experiments: While we do not have a theoretical guarantee on convergence to a stationary point, we have seen in all our experiments that the error goes down with the number of iterations.We will add these plots to our revision.

Reviewer 1:

1) Yes the fourth term is dominated by the second term and can be dropped.

2) The bound holds in expectation. In our proofs we use exponential concentration. Hence, the probability of failure can be made extremely small with little penalty. In order to get bounds on the mean squared error (MSE) as done in theorem 1 we couple the high probability bounds on MSE with worst-case bounds (which happen with exceedingly small probability) to get an expectation bound.

3) See above.

4) It is true that MMC-1 performs worse than LMaFit-A in the synthetic examples. On the synthetic datasets (section 5.1) while MMC-1 is worse than LMaFit-A it could be the case that the experimental result is very specific to the logistic transfer function and the noiseless setting used in these experiments. Results on the real dataset are a more accurate reflection of performance, and here we see that MMC-1 is competitive with LMaFit-A.

Line 333: We will add a reference here. If the noise matrix has iid sub-Gaussian entries (as assumed in this paper), then ||N|| < O(\sqrt{n}) with exponentially high probability. This allows us to guarantee that \mu_1= O(\sqrt{n}), and \mu_2=O(n). A very good reference for these results is Theorem 5.39 in Roman Vershynin's "Introduction to non-asymptotic analysis of random matrices". Reference will be included in the final version.

Line 347: We shall make this clear in our final version.

Experiments with noise: Experiments with real datasets (Section 5.2) are meant to be a better illustration of the goodness of our algorithm than synthetic experiments with noise. Also, since the NIPS community emphasizes results on real data, we tried to limit synthetic experimental results in our paper.

An apology for the way the proof is organized. We now have an updated and much well organized proof that will be uploaded in our revised version.

Reviewer 2:

Our algorithm MMC-c performs T>1 iterations. The empirical results shown in table-1 show that MMC-c gets smaller error than MMC-1. In practice we do see decreasing error with the number of iterations. We did not present these results, as the empirical results for MMC-c justify the decreasing error argument.

Use of Lipschitz constant in solving the optimization problem (14) as done in this paper does not lead to any loss of information. In contrast, using the Lipschitz constant in the gradient descent iterates shown in Section 3.1, would mean approximating the transfer function globally using a linear function, which is a poor approximation.
In experiments, unreported in this paper, we do observe that using updates (6) leads to significantly inferior performance when compared to using updates (12).

Reviewer 4:

In practice one could cross-validate for the Lipschitz constant, and choose the smallest value that gets the best performance.

Thanks for the reference. We were unaware of this reference. In the reference the authors study a collaborative filtering problem similar to ours. However, the focus of the authors is on accurate retrieval of user-wise ranking of items. In contrast we focus on user-item rating prediction. Our error bounds and experimental results are on the Frobenius norm error of the recovered matrix. Also, the method described in this paper is significantly different from the ones proposed in our paper (with regards to monotonic function learning). The reference does not provide rigorous error guarantees on the accuracy of the ranking recovered, whereas we provide rigorous error guarantees of the MSE of the proposed algorithm.

Reviewers 5, 6, 7:
Thanks for your kind comments.